# Preparation and Characterization of Polyvinylpyrrolidone/Cellulose Nanocrystals Composites

**DOI:** 10.3390/nano8121011

**Published:** 2018-12-05

**Authors:** Marina Voronova, Natalia Rubleva, Nataliya Kochkina, Andrei Afineevskii, Anatoly Zakharov, Oleg Surov

**Affiliations:** 1G.A. Krestov Institute of Solution Chemistry of the Russian Academy of Sciences, 1 Akademicheskaya St., Ivanovo 153045, Russia; miv@isc-ras.ru (M.V.); rublevanw@yandex.ru (N.R.); nek@isc-ras.ru (N.K.); agz@isc-ras.ru (A.Z.); 2Department of Physical and Colloid Chemistry, Ivanovo State University of Chemistry and Technology, 7 Sheremetevsky Prospect, Ivanovo 153000, Russia; afineevskiy@mail.ru

**Keywords:** cellulose nanocrystals, nanocomposites, self-assembly, dispersibility

## Abstract

Composite films and aerogels of polyvinylpyrrolidone/cellulose nanocrystals (PVP/CNC) were prepared by solution casting and freeze-drying, respectively. Investigations into the PVP/CNC composite films and aerogels over a wide composition range were conducted. Thermal stability, morphology, and the resulting reinforcing effect on the PVP matrix were explored. FTIR, TGA, DSC, X-ray diffraction, SEM, and tensile testing were used to examine the properties of the composites. It was revealed PVP-assisted CNC self-assembly that produces uniform CNC aggregates with a high aspect ratio (length/width). A possible model of the PVP-assisted CNC self-assembly has been considered. Dispersibility of the composite aerogels in water and some organic solvents was studied. It was shown that dispersing the composite aerogels in water resulted in stable colloidal suspensions. CNC particles size in the redispersed aqueous suspensions was near similar to the CNC particles size in never-dried CNC aqueous suspensions.

## 1. Introduction

Cellulose nanocrystals (CNC) are crystalline nano-sized rod-like particles, that can be easily produced from natural renewable cellulosic materials by controlled acid hydrolysis. When sulfuric acid is used, CNC form stable aqueous suspensions due to negatively charged sulfate ester groups at the CNC particle surface which cause electrostatic repulsion between the rod-like colloidal particles. The diameter of the anisotropic CNC particles is from 10 to 20 nm, the length is from 100 to 300 nm, depending on processing conditions and the raw material [1]. The main features that promote the use of CNC are an anisotropic particle shape, a rather large surface area, a low density, a very high modulus of elasticity and a significant reinforcing effect at a low content in a composite. As one of the most plentiful natural polymers with superb mechanical properties, excellent biodegradability and biocompatibility, nanocrystalline cellulose has been considered as a perfect reinforcing element for low-cost nanocomposites [2,3]. The properties of CNC (capacity to chemical modification of the surface hydroxyl groups, high mechanical strength, formation of a chiral nematic liquid-crystalline phase) attract considerable attention of researchers designing new functional materials [4,5,6,7,8,9].

These days the most studies on CNC composites relate to materials with low CNC content because of the challenge to prevent CNC agglomeration at high CNC content. Another issue is a choice of suitable polymer matrix providing high CNC content, nano-structural control, and interface tailoring at molecular scale level. In this regard, model systems consisting of crystalline (CNC) and amorphous (polymer matrix) components can provide a mimetic approach for research of natural phenomena [10]. In recent years, intensive efforts have been provided to design new eco-friendly polymeric materials from renewable sources as well as hybrid structures comprising nanoscale natural biodegradable and biocompatible polymers and inorganic materials [11,12].

Polyvinylpyrrolidone (PVP), a water-soluble, non-toxic, non-ionic amorphous polymer with high solubility in polar solvents, has been widely used in nanoparticles synthesis [13]. Due to the amphiphilic nature, PVP can affect morphology and nanoparticle growth by ensuring solubility in various solvents, discriminatory surface stabilization, controlled crystal growth, playing the role of a shape-control agent and facilitating the growth of specific crystal faces while preventing others [14,15,16].

PVP, with its versatile features such as a lack of toxicity, film formation, water solubility, and adhesive power, is one of the most promising polymers for nanogels preparation [17,18,19,20,21]. PVP can serve as a nanoparticle dispersant, growth modifier, surface stabilizer, hampering the agglomeration of nanoparticles through the repulsive forces that arise from its hydrophobic carbon chains interacting with each other in a solvent. PVP has been thought as a perspective material with a very good film-forming ability for potential application in production of cosmetics, printing inks, detergents, paints, coatings, shielding devices, adhesives, plastics, medicine, and pharmaceuticals [22,23,24,25].

Wu et al. [26] developed a method of polypyrrole (PPy) polymerization on CNC coated with PVP. They achieved high conductivity for the PPy/PVP/CNC composite with an excellent specific capacitance and improved cycling stability. The authors emphasize that physical adsorption of PVP onto the CNC surface plays a critical role in ensuring uniform PPy coating. PVP makes the CNC surface hydrophobic and, thus, much more appropriate for the growth of the hydrophobic PPy cover that acts as a steric stabilizer stopping further aggregation of the nanoparticles.

Going et al. [27] reported a method for achieving dispersion of the freeze-dried CNC/PVP composite in a water–methanol system using magnetic stirring and sonication. The effects of CNC loading and preparation method on dispersion, solution viscosity, and mechanical properties of electrospun PVP/CNC nanocomposites were inspected.

Huang et al. [28] produced PVP/CNC/Ag nanocomposite fibers via electrospinning and studied their properties. The authors showed that improved strength and antimicrobial characteristics of PVP/CNC/Ag electrospun composite fibers made the material a perspective candidate for a biomedical application.

Chitosan/PVP/CNC composite films were prepared by Hasan et al. [29] for the development of an efficient sustained drug release in wound dressing application. It was revealed that incorporation of CNC in films enhanced swelling, thermal, and mechanical properties and improved resistance to enzymatic degradation.

Recently, Gao and Jin [30] manufactured iridescent chiral nematic CNC/PVP nanocomposite films for fast and easy distinguishing similar organic solvents, such as halogenated hydrocarbons, skeletal isomers, and homologues. They showed that the CNC/PVP nanocomposite film could work as a sensor for the solvents recognition due to an apparent structural color change.

Here, we hypothesize that PVP adsorption onto CNC can block lateral interactions between the CNC and prevents their agglomeration in the lateral direction, i.e., hinders growth of the CNC particles width upon the concentration increase or drying of the composites. On the other hand, we suppose that freezing of CNC suspensions can align rod-like CNC particles in direction of ice crystal growth allowing formation of the CNC aggregates with a high aspect ratio. We assume also that these aggregates can be broken down easily in water and some organic solvents, providing good dispersibility of the composites.

The objective of this study is to produce and characterize PVP/CNC composite films and aerogels over a wide composition range, investigate the composites morphology and PVP-assisted CNC self-assembly as well as explore dispersibility of the PVP/CNC aerogels in water and some organic solvents.

## 2. Materials and Methods

### 2.1. Materials

Microcrystalline cellulose (MCC) (~20 micron, powder) and polyvinylpyrrolidone (powder, average Mw 40,000) (Figure 1) were purchased from Sigma-Aldrich (Saint Louis, MO, USA). Propanol, dioxane, and chloroform of analytical grade were purchased from Sigma-Aldrich Rus and were used without further purification (Moscow, Russia). Deionized water (18 mΩ, Millipore systems) was used throughout the experiment. Sulfuric acid (chemically pure, GOST 4204-77) was purchased from Chimmed (Moscow, Russia). Potassium bromide (FT-IR grade, >99%) was purchased from Sigma-Aldrich Rus (Moscow, Russia).

### 2.2. Preparation of CNC

CNC were produced from MCC using 62 wt% H_2_SO_4_ hydrolysis as described earlier [31]. The prepared CNC had a rod-like morphology with an average width of 15–20 nm and length of 100–150 nm (aspect ratio of about 7) from transmission electron microscopy (TEM) analysis (Appendix A, Appendix A).

### 2.3. Preparation of PVP/CNC Composites

PVP/CNC composites were prepared by solution casting method similar as described earlier [32]. In brief, the PVP solution was prepared by dissolving 0.5 g of PVP in 10 mL of deionized water at room temperature for 2 h under agitation. The PVP/CNC ratio was controlled by stoichiometric addition of a CNC suspension to a fixed quantity of the PVP solution followed by vigorous stirring for 1 h. The solutions were cast into glass Petri dishes and dried at ambient conditions for 24–48 h. The resulting PVP/CNC composite films were about 100 μm thick.

The aerogels based on CNC and PVP were prepared by freeze-drying of aqueous mixtures of the two components. The aqueous mixtures of CNC and PVP were prepared by ultrasonication before the samples were frozen at −40 °C for 48 h and subsequently lyophilized under reduced pressure of 6 Pa at −54 °C for 36–48 h.

The composite films or aerogels with different CNC contents (for example, 2 or 4 wt.%, etc.) were denoted as PVP/CNC-2, PVP/CNC-4, etc., or PVP/CNC-2 aero, PVP/CNC-4 aero, etc., respectively.

### 2.4. Characterization

#### 2.4.1. Determination of Size and Surface Charge of CNC Particles

The size of CNC particles was determined using an EMV-100L (Sumy, Ukraine) transmission electron microscope (TEM) (accelerating voltage of 50 kV).

The particle size distribution in aqueous and organic CNC suspensions was determined by dynamic light scattering method (DLS) (wavelength of 633 nm) using a Zetasizer Nano ZS (Malvern Instruments Ltd, Malvern, UK) device. This equipment provides measurement of particles sizes ranging from 0.3 nm to 6 μm. The measurements were carried out at temperature of 20 °C and concentration of 0.1 mg/mL in disposable polystyrene cuvettes. The obtained particle size values are the results of averaging over five consecutive measurement cycles. The value obtained in each cycle is, in turn, the result of automatic processing of 10–15 measurements. Measured by the DLS method, CNC particle sizes are the mean hydrodynamic diameters of equivalent spheres. They do not represent actual physical dimensions of the rod-like CNC particles but are valid for comparison purposes. However, the hydrodynamic diameter strongly correlates with the length of a rod-shaped particle evaluated by electron microscopy or atomic force microscopy [33,34].

The CNC particle surface charge in aqueous suspension was evaluated by measuring the ζ-potential (Zetasizer Nano ZS, Malvern Instruments Ltd, Malvern, UK).

#### 2.4.2. Scanning Electron Microscopy (SEM)

The investigations of the surface morphology of the composites were carried out using a VEGA3 TESCAN instrument (Brno, Czech Republic) (accelerating voltage of 5 kV, a chamber pressure of about 9 × 10^−3^ Pa, contrast in a secondary electron mode).

#### 2.4.3. Fourier Transform Infrared Spectroscopy (FTIR)

The FTIR measurements of the samples (pressed tablets with KBr) were carried out at room temperature using a VERTEX 80v FTIR spectrometer (Bruker, Ettlingen, Germany) in the range of 4000 to 400 cm^−1^.

#### 2.4.4. X-ray Diffraction Analysis

The X-ray diffraction analysis was performed using a Bruker D8 Advance powder diffractometer (Bruker BioSpin GmbH, Rheinstetten, Germany) according to the Bragg–Brentano scheme with Cu-*K*α radiation (λ = 0.1542 nm).

CNC crystallite sizes were estimated using the Scherrer equation [35]:*L* = *Kλ*/*ß*_1/2_*Cosθ*,(1)
where, *L* is the crystal dimension perpendicular to the diffracting planes with *hkl* Miller indices, *λ* is the wavelength of X-ray radiation, *ß*_1/2_ is the full width of the diffraction peak measured at half maximum height (FWHM), *θ* is the diffraction angle, and *K* = 0.94.

#### 2.4.5. Thermogravimetric Analysis (TGA)

Thermogravimetric analysis was performed on a TG 209 F1 (Netzsch Gerätebau GmbH, Selb, Germany) at a heating rate of 10 K min^−1^ in an atmosphere of dry argon at a flow rate of 30 mL min^−1^. 

#### 2.4.6. Differential Scanning Calorimetry (DSC)

The DSC measurements were performed on a DSC 204 F1 (Netzsch Gerätebau GmbH, Selb, Germany) in an atmosphere of ultra pure grade dry argon at a flow rate of 15 mL min^−1^ and a heating rate of 10 K min^−1^ using standard aluminum crucibles.

#### 2.4.7. Mechanical Properties

The tensile strength and elongation at break were measured using an IR 5046-5 tensile-testing machine (Ivanovo, Russia) at a loading rate of 1 mm/min and at room temperature with a solid clamp in the tension mode [36]. Four specimens with the dimensions of 15 mm (length) × 5 mm (width) × 0.1 mm (thickness) were used for each sample group. The stress and strain values were calculated from the machine-recorded force and displacement based on the initial cross-section area and the original gauge length (10 mm) of each sample, respectively. The Young’s modulus for each sample was calculated from the initial linear portion of the stress-strain curves through a linear regression analysis. The obtained values of the Young’s modulus were within ±10%, while the stress and the elongation at break fluctuated in the range of ±15%. Glycerol was added (2% *v*/*v*) to the CNC solution to enhance the elasticity of the films, and this solution was further used to dissolve PVP.

#### 2.4.8. Water Uptake

Isotherms of water adsorption and desorption upon films and aerogels of the PVP/CNC composites were obtained. The sorption experiment was conducted in a desiccator at temperature of 25 °C under controlled humidity (aqueous solutions of sulfuric acid of a certain concentration). The water uptake was determined gravimetrically.

## 3. Results and Discussion

### 3.1. FTIR Analysis

Figure 2 shows FTIR spectra of PVP/CNC composite films. The peaks located at about 1291, 1426, 1666, and 2953 cm^−1^ were assigned to the stretching vibrations of C–N, C=C, C=O, and C–H of PVP, respectively [13]. The appearance of a broad band at about 3440 cm^−1^ suggests that PVP contains adsorbed water. The large bands near 2900 cm^−1^ and at 3300–3500 cm^−1^ assigned respectively to the C–H and O–H stretching vibrations of cellulose, were found for all the characterized materials. The characteristic bands at 1060 and 1030 cm^−^^1^ and at 1165, corresponded to the C–OH stretching vibrations of the cellulose secondary and primary alcohols and to the asymmetric ring breathing mode of cellulose, respectively [37]. The stretching vibration of the cellulose C–O–C bond was identified at 1110 cm^−^^1^ and 1050 cm^−^^1^. The FTIR spectra of all the PVP/CNC composites demonstrated an intense wide band at about 3440 cm^−1^ related to adsorbed water. As the CNC content in the composites increased, almost no changes in the positions of all these peaks were noted. Thus, the FTIR spectra indicated that the molecular interactions between CNC and PVP are weak (see Section 3.9). However, the data obtained [38] suggest that in complexation reactions with substances of a different chemical nature, PVP participates in hydrated state through bound water. In this case, water can stabilize the resulting intermolecular complexes. It is obvious that the PVP interaction with water occurs on account of hydrogen bonding. It is believed that a PVP unit forms hydrogen bonds with two water molecules through lone-electron pairs of the oxygen atom of the carbonyl group of the pyrrolidone heterocycle [38].

### 3.2. Water Adsorption upon the PVP/CNC Composites

Water sorption is one of the major problems limiting applications for CNC based materials. Water uptake affects mechanical properties, chemical and dimensional stabilities. On the other hand, the water absorption capacity together with the biodegradability are among the most important properties for the practical applications and post-use of the biodegradable materials. There is a plethora of studies that have investigated the moisture sorption and retention by CNC reinforced polymer composites [39,40,41,42,43,44]. Absorbed water molecules can be accumulated in three regions within a CNC-based nanocomposite: on the CNC, in a polymer matrix, and at the polymer/CNC interface [45]. The accessible hydroxyl groups on the CNC particles surface are commonly considered as the initial site of water sorption. However, for polymer composites, the CNC networks can decrease water sorption as compared to the neat polymer matrices that readily adsorb water [46].

Isotherms of water adsorption and desorption upon films and aerogels of the PVP/CNC composites were obtained. The adsorption experiment was carried out as described earlier [32]. A detailed analysis of the isotherms is depicted in Figure 3. Obviously, water uptake increases with a relative pressure increase. Meanwhile, the water uptake on the composites decreases gradually with the CNC content increase. Water uptake by the aerogels is very similar to the uptake by the films in spite of their different densities (see Section 3.3).

As PVP is a hydrophilic polymer, it has a great tendency to adsorb water. Evidently, the water uptake by the composites does not depend on either morphology (film or aerogel) or density of the sample, but is determined only by the PVP content in the composite. The molecular interactions between CNC and PVP are weak, but they may interact with each other through bound water. Apparently, the reduced water binding capacity of the PVP/CNC composites as compared to the neat PVP can be related to the increase in the intramolecular CNC–CNC and PVP–PVP interactions (see Section 3.5 and Section 3.9).

### 3.3. Density of the PVP/CNC Composite Films

Figure 4 shows the composites density versus the CNC content.

The density was determined by measuring the samples geometry and dividing the weight by the volume. The reported results are the average values obtained from at least three different samples. It can be seen that the dependence of the film density on the CNC content is of complex nature: the film density is maximal at the CNC content of about 5–10 wt% and minimal at the CNC content of about 55–60 wt%. However, the aerogel density gradually decreases with the increase in the CNC content in the composite. 

The strange density behavior of the composite films deserves a special attention and some speculation. If we assume that in the composite films the PVP molecules are in a coil (or globule) conformation, then their shape can be considered to be spherical (see Section 3.9). The phase diagram for packing of mixtures of spheres and rods is very complex because it depends on the stoichiometry, the rod aspect ratio, and size ratio of the rod and the sphere (*R*_sphere_/*R*_rod_). Simulations predict the densest packing for binary structure of spheres and rods (of 2:1 stoichiometry) at an appropriate *R*_sphere_/*R*_rod_ range, that exceeds the densities of the demixed phases (rods and spheres) [47]. However, at intermediate volume fractions, simulations predict phase separation into rods and spheres. Moreover, depending on volume fractions, size ratio or aspect ratio, packing of phase-separated rods and spheres may be more effective than packing of binary structure of spheres and rods. Thus, we observe a complex density behavior of the composite films as a result of the packing efficiency of rod-like CNC particles and spherical PVP molecules.

### 3.4. TG Analysis

The TG and derived differential TG (DTG) curves of the PVP/CNC composite films are shown in Appendix A, respectively. The thermal parameters, including the maximum thermal degradation temperature (*T*_max_) and the onset thermal degradation temperature (*T*_on_) are summarized in Table 1.

The PVP-based composites with a small CNC content have one main characteristic mass loss region (excluding the initial region of water desorption). On the thermograms of the composites with the CNC content of about 7% or more, two main mass loss regions are observed. The water content in the samples gradually decreases with the growing CNC content, and for the neat CNC it equals 3.0%.

The first main degradation range is located between ~230 and 400 °C and is attributed to pyrolysis of the CNC catalyzed by acid sulfate groups [48]. The second stage mass loss occurs at about 400 °C and involves decomposition of PVP and CNC carbonaceous matter [31].

### 3.5. DSC Analysis

Before the DSC scanning, all the samples were first equilibrated at 20 °C and heated to 200 °C at a heating rate of 10 K min^−1^, and then cooled down to 20 °C at the same rate (Appendix A). The glass transition temperatures (*T*g) for all the samples determined under heating and cooling are reported in Table 2. The results of DSC analysis of the PVP/CNC films show that *T*g tends to shift to higher values with increasing the CNC content in the composite. This higher *T*g value can point out that the association of PVP molecules is enhanced by the CNC presence. A notable increase in *T*g may be ascribed to the macromolecular confinement provided by the CNC surfaces [49]. On the other hand, the nanoconfinement effects may facilitate significant enhancement of mechanical properties.

However, analyzing the data on density, as well as the results of TG and DSC analysis, one can observe an interesting feature. Composite films with a CNC content of about 5–10% are characterized by the maximum density, the maximum temperature of water desorption and the minimum *T*g under heating. Apparently, this is due to the firmly bound water, which is not completely removed from PVP even under heating up to 250 °C [38]. Indeed, the strongly bound water can act as a plasticizer and increase mobility of the polymer chains (the *T*g decreases).

### 3.6. Mechanical Properties

Appendix A shows stress-strain curves of the PVP/CNC composite films. The values of ultimate tensile strength (*σ*_max_), Young’s modulus (*E*), and elongation at break (*ε*_b_) are summarized in Table 3.

Upon the addition of CNC, *σ*_max_ of the composites increases, indicating a reinforcing effect of the CNC. However, when CNC is added, *ε*_b_ decreases gradually, which suggests that the composites become brittle compared with the neat PVP. As the CNC content grows, *σ*_max_ and *E* increase and reach maxima at the CNC content of 29.2 wt.%. However, upon further increasing of the CNC content, *σ*_max_ and *E* decrease. Therefore, a CNC content of about 30 wt.% is optimal for the composite films based on the *σ*_max_ and *E* values.

Various modeling approaches have investigated the effect of CNC fillers within polymer matrices on effective nanocomposite properties. Typically, Halpin–Tsai, Halpin–Kardos, and percolation approaches have been used to understand the reinforcing effect of low concentration CNC in low-modulus polymers [2]. Experimentally obtained tensile modulus could be compared with theoretical predictions by using a modified Halpin–Tsai model as [50]:*E*_c_/*E*_m_ = (3/8)·(1 + 2*ρη*_L_*V*_CNC_)/(1 − *η*_L_*V*_CNC_) + (5/8)·(1 + 2*η*_T_*V*_CNC_)/(1 − *η*_T_*V*_CNC_),*η*_L_ = (*E*_r_ − 1)/(*E*_r_ + 2*ρ*),*η*_T_ = (*E*_r_ − 1)/(*E*_r_ + 2),(2)
where *E*c and *E*m are the Young’s modulus of the composite and matrix, respectively; *ρ* is the CNC aspect ratio; *E*_r_ is the ratio between the Young’s modulus of CNC and the matrix; *V*_CNC_ is CNC volume fraction in the composite.

Based on the weight percentage (W_CNC_), one can estimate the corresponding volume fraction *V*_CNC_ by knowing the density of CNC and PVP as:*V*_CNC_ = (*W*_CNC_/*d*_CNC_)/(*W*_CNC_/*d*_CNC_ + (1 − *W*_CNC_)/*d*_PVP_)(3)

The Young’s modulus of CNC nanoparticles has been set at 105 GPa; the CNC aspect ratio has been assumed as 7; the densities for CNC and PVP have been assumed as 1.46 and 1.2 g/cm^3^, respectively.

Figure 5 compares the experimentally obtained data and predicted Young’s modulus values according to random Halpin–Tsai equation. The Halpin–Tsai model tends to lower predictions for high modulus ratio systems *E*_r_, that is common in nanocomposites based on rubbery matrices [2]. These predicted values are not accurate for composites with aligned stiff short fillers. Besides inaccuracy at high modulus ratio, the Halpin–Tsai equation has no information about filler anisotropy or about the quality of the interface. However, a stiffening of the interphase region might be very important and useful for stress transfer from the matrix to the CNC. Thus, an observation that experiments exceed the Halpin–Tsai model is the expected result in case of the CNC filler and a low-modulus polymer.

The observed experimental results exceeding Halpin–Tsai model can be explained in terms of percolation concept in CNC based composites. The concept of percolation is related with the development of a connected network in a multiphase system. The percolating phase reaches the percolation threshold, i.e., a connected network is formed, when the concentration of the percolating phase increases. Some effective properties increase rapidly and dramatically for concentrations above the percolation threshold. In nanocomposites, percolation is most dramatic for high modulus ratio systems, and the percolation threshold can be shifted to very low concentrations by using high aspect ratio nanofillers [51].

The Young’s modulus and tensile strength of CNC reinforced polymer matrix composites can be compared to other material systems via ‘Ashby plots’. According to Gibson–Ashby theory, Young’s modulus and strength of a solid are proportional to the relative density [52]. Possible reasons for poor enough mechanical properties in CNC based composites are CNC aggregation, low interfacial properties of CNC/matrix, low particle aspect ratio, and low CNC alignment [2]. The experimental data obtained are in good agreement with the density data, according to which the density of composite films decreases sharply with a CNC content of more than 30% (Figure 4a).

### 3.7. X-ray Diffraction Analysis

The X-ray diffraction patterns of the PVP/CNC films are shown in Figure 6.

As PVP is an amorphous polymer, the diffraction peak arising at about 2*θ* = 22.9 is attributed to the (200) plane of cellulose I_ß_, whereas the two overlapped weaker diffractions at 2*θ* close to 16.6° and 14.8° are assigned to the (110) and (1–10) lattice planes of cellulose I_ß_ [53]. The intensity of the diffraction peaks for CNC becomes more pronounced when the CNC content in the composite is increased.

The powder diffraction peaks were deconvoluated with the Pseudo-Voigt function, assuming a linear background [54]. The average dimensions of the elementary crystallites in the diffracting planes were evaluated by applying the Scherrer equation (1). From these values, we suppose the crystallites have a square cross-sectional geometry with widths of about 2.6 nm and a diagonal of 3.9 nm (Table 4, Figure 7).

### 3.8. Morphologies of the PVP/CNC Composites

SEM images were recorded to analyze the microstructure of the PVP/CNC composite films and aerogels (Figure 8, Figure 9 and Appendix A).

The cross-sectional SEM image of the neat PVP film showed a smooth morphology (Figure 8a). With the increase in the CNC content in the composite films, the fractured surface became rougher due to the CNC particles aggregation (Figure 8b–f). Separate CNC particles in the films were not observed. Analyzing SEM images of the composite aerogels, we unexpectedly found that in a certain composition range (the CNC content from about 5 to 30 wt.%), the individual CNC particles became visible (Figure 9b–e).

Using the SEM images, we estimated the CNC particle size in the composite aerogels. For this purpose, we measured the widths and lengths of a set of 100 particles manually (Appendix A). The analysis results are shown as histograms in Figure 10.

The analysis of the images shows that the widths of the CNC particles in the composite aerogels are about 55–65 nm and do not depend on the composition. Meanwhile, the lengths of the CNC particles in the aerogels are much larger than the lengths of the CNC particles in the initial suspension, these composites were produced from. In addition, as the CNC content in the composite grows, the particles lengths increase gradually and exceed 1 micron for the composition with a CNC content of 28.9 wt.%. The fraction of particles with the lengths exceeding 600 nm also increases.

It is reasonable to assume that such an increase in the CNC particles size is due to their aggregation. It is well known that CNC strongly agglomerate to stack up in parallel giving an apparent increase in particle width [54]. However, it should be noted that in this case their lengths increase, while the widths remain unchanged. Evidently, PVP assists CNC self-assembly which produces reasonably uniform CNC aggregates with a high aspect ratio (length/width). The aspect ratio of the CNC aggregates increase as the CNC content grows from about 5 to 30 wt.% because the lengths increase up to fourfold, while the widths remain constant. Apparently, here we observe the result of a synergistic effect of PVP adsorption onto CNC particles and ice-templating under freeze-drying [56,57]. As inter-chain hydrogen bonding is prevailing within the (200) plane for cellulose I_ß_ (the so-called ”hydrogen-bonded” plane) [2,58,59], one can suppose that adsorbed PVP blocks the possibility of forming lateral bonds between the CNC particles and, thus, prevents their aggregation. Freezing of CNC suspensions can align rod-like particles because the particles can be subjected to an elongation action as the speed of ice crystals growth along the length differs from their growth across the width [60,61]. Moreover, some tribology findings highlight that polymer adsorption on CNC surface together with the CNC surface hydration can act as a lubrication and reduce a friction in the CNC contacts considerably [46].

### 3.9. PVP Adsorption onto CNC Particles

PVP as a linear amorphous polymer with a large heterocyclic unit in the polymer chain is characterized by a coil-globule conformational transition in solution, which is caused by the interaction of distant units in the chain. If the process is driven by units repulsion, the coil swells. If the monomer units are attracted to each other, the polymer chain condenses onto itself into a dense conformation—the so-called polymer globule. It is thought that at higher temperatures or in a good solvent, mutual repulsion of the polymer units prevails, and the coil swells due to the excluded volume effect. At low temperatures or in a poor solvent, the chains are mostly attracted to each other, and the coil collapses into a globule. A very small attraction between the polymer chain units is usually enough to cause the polymer to condense into a globule. In polymers with flexible chains (as PVP), the coil-globule transition is a continuous process.

Figure 11 shows the size distribution of PVP particles depending on the concentration in solution and their surface charge determined by the ζ-potential value. With an increase in the PVP concentration from 0.05 to 7%, the fraction of particles with the size of 60–70 nm disappears, while the fraction of particles with sizes less than 10 nm grows. At increased PVP concentrations, the average size of these particles is reduced approximately to 3–6 nm. The experimental values of the PVP particle size obtained by dynamic light scattering are in good agreement with the estimated ones. Thus, the length of the expanded polymer chain of PVP with the molecular mass of 40,000 is about 50 nm, and the radius of gyration of the coil, *R*_g_, varies from 1 to 7 nm that can be calculated from the known relations
*R*_g_ = 0.0195·*M*^0.55^,(4)
where *M* is the molecular weight of PVP [62];
*R*_g_ = (*L*·*N*/6) ^0.5^,(5)
where *L* is the bond length in the polymer chain, nm; *N* is the number of the bonds [63].

It is seen that a concentration increase reduces the fraction of expanded polymer chains, and the degree of coil swelling becomes lower, which leads to the coil contraction and decrease in its size (Figure 11a). The PVP coil contraction is accompanied by an increase in the absolute values of the ζ-potential (Figure 11b), which agrees well with the results of quantum-chemical calculations [38], according to which the negative charge is caused by the formation of H-bonded complexes of the pyrrolidone ring with water molecules and should become lower (more positive) at dehydration. The PVP adsorption onto CNC particles is accompanied by growth in the ζ-potential absolute value, but does not significantly affect the CNC particle size (Figure 12a,b).

It is usually believed that PVP molecules tend to be adsorbed onto nanoparticles in a linear open-chain-like pattern as opposed to a compact coiled structure, i.e., the molecular coils are flattened at the surface [64]. Nevertheless, the surface features of the solid adsorbents can affect the properties of the adsorbed layer. Hydrogen bonding, electrostatic attraction, or hydrophobic interactions can highly influence the adsorption behavior [63]. One can hypothesize that strong adsorption due to the high affinity of PVP for the adsorbent surface causes the PVP molecules to be adsorbed in an open-chain-like pattern. On the other hand, if the PVP does not interact well with the adsorbent surface, the PVP molecules tend to be adsorbed in more compact coiled shaped structures.

For example, atomic force microscopy was used by Verma et al. for research the interaction of PVP with ibuprofen [65]. The obtained AFM images showed that PVP tended to be adsorbed in a compact coiled structures, demonstrating a lack of affinity for the ibuprofen surface. In addition, the authors observed patchy preferential adsorption of PVP on one face of the ibuprofen crystal as compared to the other.

Kotel’nikova et al. [66] studied the effects of PVP molecular weight and aqueous solution concentration of PVP on adsorption upon microcrystalline cellulose. The authors observed a noticeable increase in the amount of adsorbed PVP with an increase in its molecular weight, and at higher solution concentration and temperature. The authors classified the obtained two-step S-shaped isotherm as an isotherm of type IV and drew a conclusion about the polymolecular character of the PVP adsorption upon the porous surface of microcrystalline cellulose. Besides, the authors noted that PVP with a larger molecular weight is both more easily adsorbed and desorbed. Apparently, this indicates that the polymer is adsorbed in compact globular structures rather than in a flattened conformation as longer linear chains with a larger molecular mass should bond to the adsorbent surface more firmly.

Hirai and Yakura [62] studied PVP adsorption on palladium (Pd) nanoparticles in methanol. They determined that the thickness of PVP adsorbed layer on Pd nanoparticles as well as the PVP radius of gyration increased proportionally to *Mw*^0.55^, where *Mw* was an averaged molecular weight of PVP. According to the authors, the adsorbed layer thickness is almost completely determined by the particular segmental sequence of the adsorbed polymer chain which takes the conformation similar to that of the free polymer in solution. Consequently, a polymer coiled conformation in solution should lead to the polymer adsorption in a globule-like structure.

Thus, analyzing the data obtained, one can suppose that PVP is adsorbed onto CNC in a globule-like structure rather than in a flattened conformation and due to electrostatic attraction rather than via hydrogen bonding. Nevertheless, the situation can be more complicated because of PVP strong binding via hydrogen bonding to the (200) facet of the CNC particle which may result in a flattened conformation of the adsorbed PVP molecules. Computer simulation could reveal and prove the appropriate mechanism of PVP adsorption onto different facets of CNC particles. However, this will be the aim of our further research.

### 3.10. Dispersibility of Freeze-Dried PVP/CNC Composites in Water and Some Organic Solvents

As a PVP molecule contains a strongly hydrophilic pyrrolidone moiety and a considerable hydrophobic alkyl group, adsorbed PVP can prevent aggregation of nanoparticles through repulsive forces which arise from the hydrophobic carbon chains. PVP can provide a steric hindrance that will allow the dried PVP/CNC to be readily redispersed in aqueous systems [27,67,68].

Thus, PVP adsorption onto the CNC surface can lead to a steric hindrance preventing hydrogen bonding between the CNC particles, which can impart good redispersibility of dried PVP/CNC in water. To test whether irreversible CNC particle aggregation occurred after drying, we redispersed freeze-dried PVP/CNC composites in water and some organic solvents. The redispersibility of the dried PVP/CNC aerogels in water was evaluated in terms of particle-size distribution (Figure 13 and Figure 14) and colloidal stability of the suspensions (Appendix A).

Appendix A displays the photographs showing the suspension stability for all the PVP/CNC samples. The photographs of the suspensions in the testing vials are taken some time after ultrasonication. All the CNC suspensions are turbid and no sedimentation can be observed at the bottom of the vials at the initial stage. As time passes, lots of sedimentation is flocculated in propanol, chloroform, and dioxane, whereas aqueous suspensions retain their good stability even a month later.

Figure 13a and Figure 14 show particle size distribution of the PVP/CNC aerogels in water and propanol, respectively. Figure 13b demonstrates surface charge of PVP coated CNC particles after redispersion of the PVP/CNC aerogels in water.

Figure 12 and Figure 13 show that the particle size and surface charge after redispersing the PVP/CNC aerogels in water remain practically unchanged compared to the initial values for the aqueous never-dried CNC in the presence of PVP. The redispersed aqueous suspensions demonstrate high colloidal stability after 1 month of holding (Appendix A).

The redispersibility of PVP/CNC aerogels in propanol is not as effective. If the PVP content in the composite is less than 60%, aggregates of 1 μm in size are formed (Figure 14). However, the redispersed suspensions with a high PVP content are stable in propanol for 10 days (Appendix A).

In chloroform and dioxane, the redispersed particles of PVP/CNC aerogels form aggregates larger than 3 μm. However, these suspensions are stable for some time. It is worth noting that the redispersed suspensions behave differently in chloroform and dioxane. If the former suspension is most stable at minimum and maximum PVP dosages, the latter, on the contrary, is most unstable under the same conditions (Appendix A).

These results testify that the enhanced redispersibility of dried CNC/PVP is not due to the electrostatic mechanism only, rather is specified by both the electrostatic repulsion and the steric hindrance from the PVP that are adsorbed onto the surface of CNC particles [69].

Appendix A illustrates PVP particles size distribution in propanol and chloroform (0.2 wt.% concentration). It is evident that compared to aqueous solutions, the hydrodynamic diameters of PVP particles in propanol and especially in chloroform tend to increase, probably due to the polymer coils extension into linear chains. It would be interesting to reveal the relationship of PVP conformations in various solvents (linear chain, coil, condensed globula) with the possibility to stabilize CNC suspensions in these solvents. However, this aim is well beyond the scope of the article and will be achieved in the future work. The specific application of such a system could involve the development of an effective drug delivery system. The aim of the further research is to enhance physiochemical properties as well as bioavailability of poorly water-soluble drugs, through preparation of freeze-dried optimized formulations based on PVP and CNC as hydrophilic carriers.

## 4. Conclusions

The work describes the preparation of PVP/CNC composite films and aerogels with extensive characterization. The composites morphology and PVP-assisted CNC self-assembly are explored. The redispersibility of the aerogels prepared by freeze-drying is investigated in detail. In this work, we have discovered PVP-assisted CNC self-assembly which produces CNC aggregates with a high aspect ratio. It was revealed that the particle size of the CNC aggregates after redispersing in water returned to the initial values for the aqueous never-dried CNC. The redispersed aqueous suspensions demonstrated high colloidal stability after 1 month of holding. It was shown that the redispersibility of CNC in an organic solvent (propanol, dioxane, chloroform) could be significantly improved using a PVP content of 50 wt.%. 

## Figures and Tables

**Figure 1 nanomaterials-08-01011-f001:**
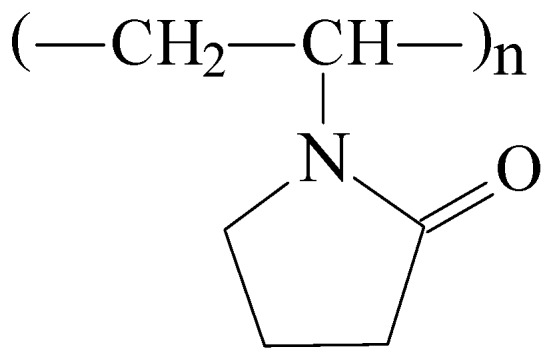
Structural formula of polyvinylpyrrolidone.

**Figure 2 nanomaterials-08-01011-f002:**
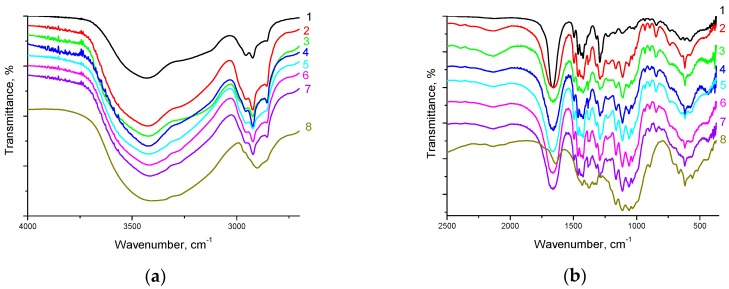
FTIR spectra of the PVP/CNC composite films: 1—neat PVP; 2—PVP/CNC-4.6; 3—PVP/CNC-10.9; 4—PVP/CNC-16.3; 5—PVP/CNC-19.6; 6—PVP/CNC-28.9; 7—PVP/CNC-37.9; 8—neat CNC: (**a**) in the range of wavenumbers of 4000–2500 cm^−1^; (**b**) in the range of wavenumbers of 2500–500 cm^−1^. PVP, polyvinylpyrrolidone; CNC, cellulose nanocrystals.

**Figure 3 nanomaterials-08-01011-f003:**
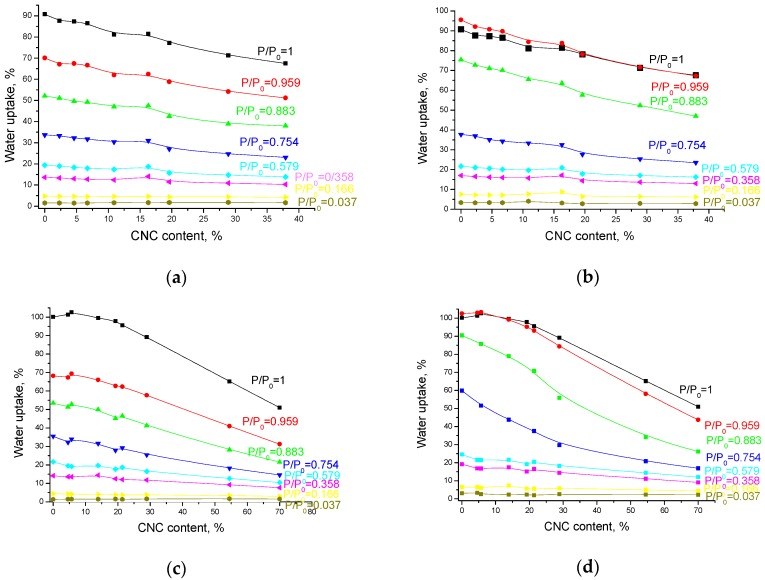
Equilibrium water content in the composite films (**a**,**b**) and aerogels (**c**,**d**) at water adsorption (**a**,**c**) and desorption (**b**,**d**) depending on the CNC content (P/P_0_ is a relative pressure of water vapor).

**Figure 4 nanomaterials-08-01011-f004:**
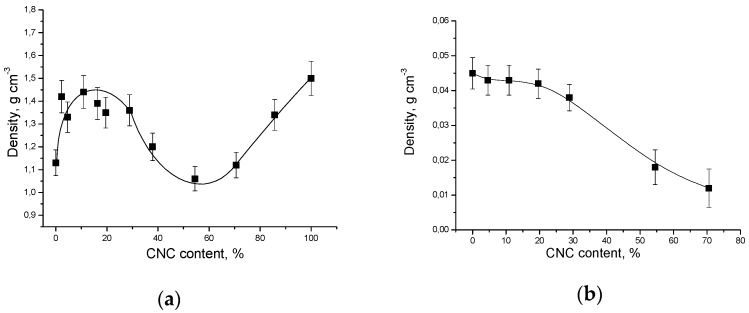
Density of PVP/CNC composites as a function of the CNC content: (**a**) films; (**b**) aerogels. The error bars are the standard deviations in repeated experiments.

**Figure 5 nanomaterials-08-01011-f005:**
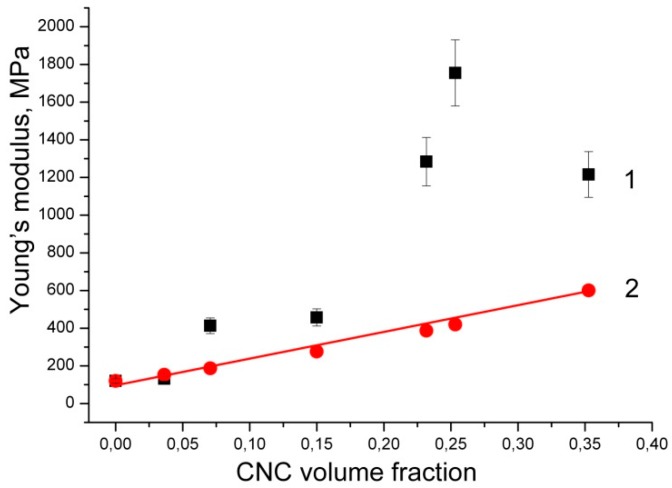
Experimental data (1) and fitting results according to Halpin–Tsai model (2). The error bars are the standard deviations in repeated experiments.

**Figure 6 nanomaterials-08-01011-f006:**
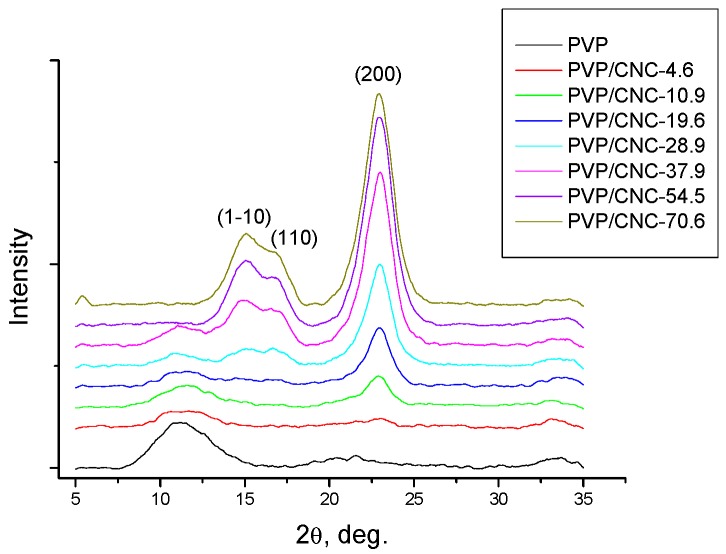
X-ray diffraction patterns of the PVP/CNC composite films.

**Figure 7 nanomaterials-08-01011-f007:**
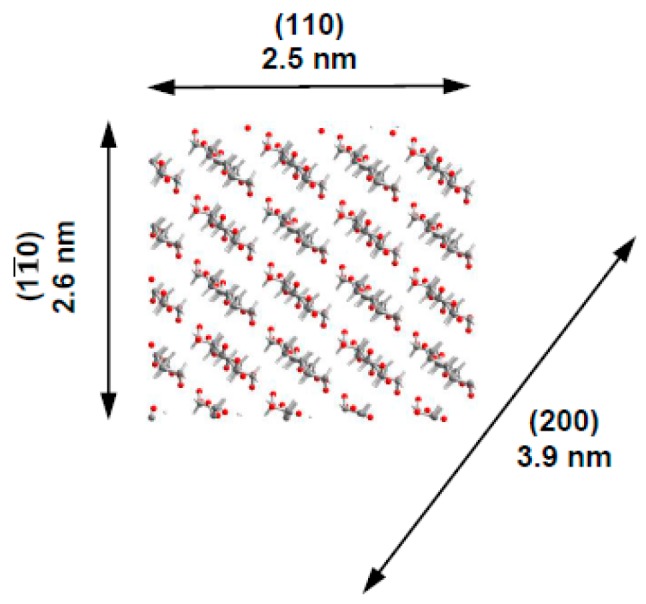
Cross-section of elementary crystallites in CNC particles. Projection of cellulose molecules is clearly visible. The indexation of corresponding lattice planes is described according to the monoclinic unit cell of allomorph I_ß_ [55].

**Figure 8 nanomaterials-08-01011-f008:**
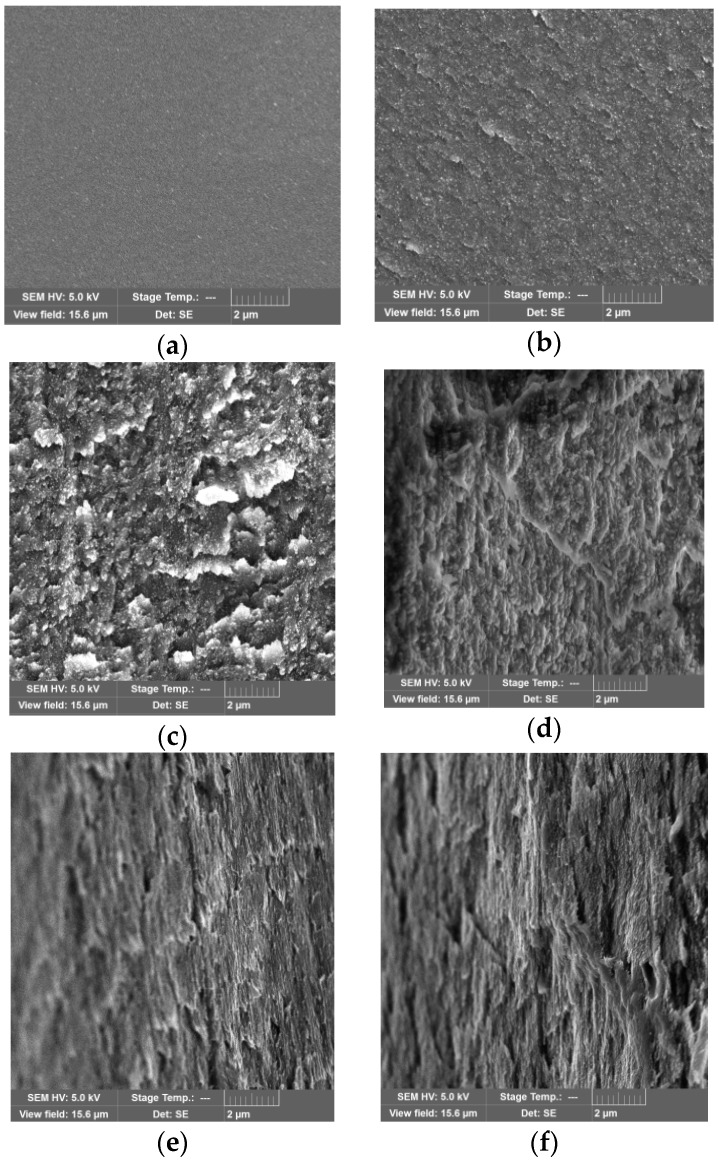
Cross-sectional SEM images of the neat PVP film (**a**) and the PVP/CNC composite films with CNC content (wt.%) of: 4.6 (**b**); 16.3 (**c**); 28.9 (**d**); 54.5 (**e**); 85.7 (**f**), respectively. The scale bar is 2 μm.

**Figure 9 nanomaterials-08-01011-f009:**
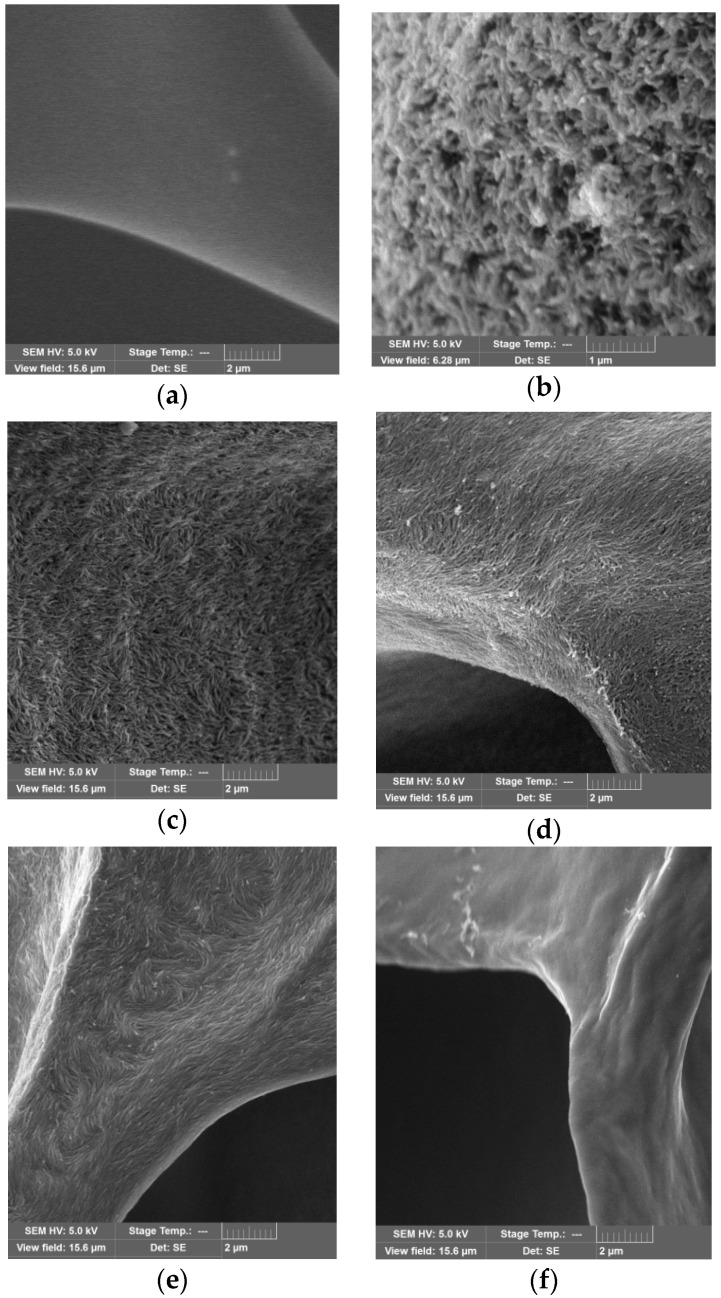
SEM images of the neat PVP aerogel (**a**) and the PVP/CNC composite aerogels with CNC content (wt.%) of: 4.6 (**b**); 10.9 (**c**); 19.6 (**d**); 28.9 (**e**); 70.6 (**f**), respectively. Scales: 2 µm (**a**,**c**,**d**,**e**,**f**); 1 µm (**b**).

**Figure 10 nanomaterials-08-01011-f010:**
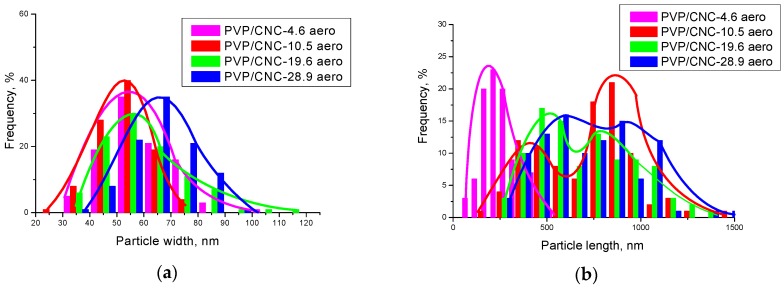
The CNC particle size distribution in the composite aerogels across the widths (**a**) and along the lengths (**b**).

**Figure 11 nanomaterials-08-01011-f011:**
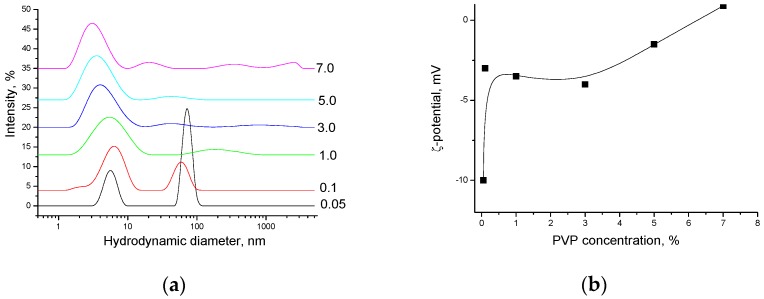
PVP particles size distribution (the curves are marked by PVP concentration in wt.%) (**a**) and the PVP particles surface charge in aqueous solutions (**b**).

**Figure 12 nanomaterials-08-01011-f012:**
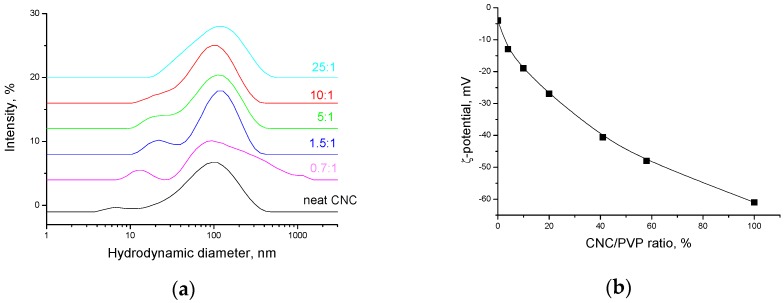
CNC particles size distribution in the presence of PVP (the curves are marked by the PVP/CNC concentration ratio) (**a**) and the CNC particles surface charge in aqueous suspensions (the CNC concentration of 0.07 wt.%) in the presence of PVP (**b**).

**Figure 13 nanomaterials-08-01011-f013:**
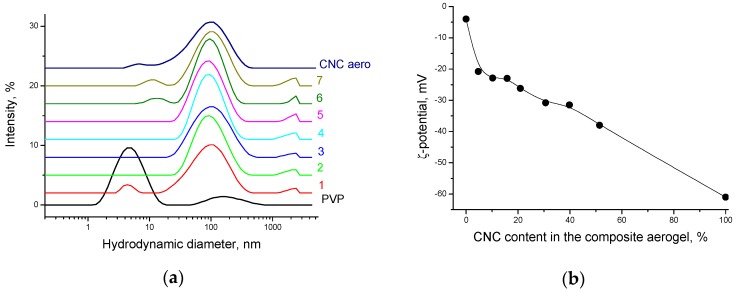
(**a**) Particle size distribution of the PVP/CNC aerogels in water (for comparison, the curves for the neat PVP and CNC are shown): 1—PVP/CNC-4.6 aero; 2—PVP/CNC-10.9 aero; 3—PVP/CNC-16.3 aero; 4—PVP/CNC-19.6 aero; 5—PVP/CNC-28.9 aero; 6—PVP/CNC-37.9 aero; 7—PVP/CNC-54.5 aero. (**b**) Surface charge of PVP coated CNC particles after redispersion of the PVP/CNC aerogels in water.

**Figure 14 nanomaterials-08-01011-f014:**
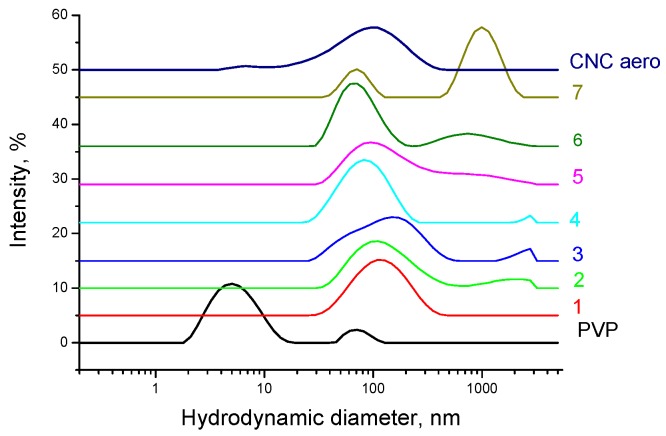
Particle size distribution of the PVP/CNC aerogels in propanol (for comparison, the curves for the neat PVP and CNC are shown): 1—PVP/CNC-4.6 aero; 2—PVP/CNC-10.9 aero; 3—PVP/CNC-16.3 aero; 4—PVP/CNC-19.6 aero; 5—PVP/CNC-28.9 aero; 6—PVP/CNC-37.9 aero; 7—PVP/CNC-54.5 aero.

**Table 1 nanomaterials-08-01011-t001:** Results of TG analysis of the PVP/CNC composite films.

Sample	Initial Mass Loss			First Degradation Stage			Second Degradation Stage			Total Mass Loss, %
	*T*_on_, °C	*T*_max_, °C	Mass Loss, %	*T*_on_, °C	*T*_max_, °C	Mass Loss, %	*T*_on_, °C	*T*_max_, °C	Mass Loss, %	
PVP	-	76.4	14.6	-	-	-	405.4	432.3	80.9	95.5
PVP/СNС-2.3	-	77.9	15.1	-	-	-	403.3	432.6	78.4	93.5
PVP/СNС-6.7	-	80.6	13.7	240.1	264.5	3.9	402.9	433.8	76.6	94.2
PVP/СNС-10.9	-	69.0	12.8	239.5	259.8	6.2	402.9	434.9	73.7	92.7
PVP/СNС-19.6	-	72.8	12.2	232.6	252.1	11.7	403.3	436.6	64.0	87.9
PVP/СNС-28.9	-	66.6	11.0	229.8	246.3	16.0	401.9	434.1	59.9	86.9
PVP/СNС-37.9	-	64.8	10.6	226.4	242.4	19.6	398.2	430.4	53.8	84.0
CNC	63.0	82.2	3.0	154.5	169.8	37.8	334.1	376.4	30.1	70.9

**Table 2 nanomaterials-08-01011-t002:** Results of DSC analysis of the PVP/CNC composite films.

Sample	*T*_g_, °C	
	Heating	Cooling
PVP	180.2	166.7
PVP/CNC-4.6	176.5	173.9
PVP/CNC-10.6	178.3	174.7
PVP/CNC-19.9	184.2	177.7
PVP/CNC-28.9	198.5	186.4
PVP/CNC-37.9	198.6	184.1

**Table 3 nanomaterials-08-01011-t003:** Tensile properties of the PVP/CNC composite films.

Sample	Ultimate Tensile Strength (*σ*_max_), MPa	Elongation at Break (*ε*_b_), %	Young’s Modulus (*E*), MPa
PVP	6.6	5.5	120
PVP/CNC-4.4	9.7	7.4	132
PVP/CNC-8.4	17.2	4.2	413
PVP/CNC-17.7	31.8	6.9	457
PVP/CNC-26.8	38.4	3.0	1284
PVP/CNC-29.2	43.8	2.5	1755
PVP/CNC-39.9	25.3	2.1	1216

**Table 4 nanomaterials-08-01011-t004:** CNC crystallite sizes (nm) in the PVP/CNC composite films.

Sample	Crystallographic Plane		
	(1–10)	(110)	(200)
PVP/CNC-28.9	2.96	2.62	3.74
PVP/CNC-37.9	2.15	2.53	4.19
PVP/CNC-54.5	2.67	2.37	3.81
PVP/CNC-70.6	2.48	2.42	3.74

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
