# Peer review of "Preparation and Characterization of Polyvinylpyrrolidone/Cellulose Nanocrystals Composites"

_nanomaterials, 2018, doi:10.3390/nano8121011_

Reviewer 1 Report

The paper can be accepted as is. 

Author Response

English language and style are improved.

Reviewer 2 Report

The manuscript deals with polyvinylpyrrolidone/cellulose nanocrystals based nanocomposites from the preparation to characterization. The topic is interesting and there are new aspects worth of being published after revision mainly devoted to improve the impact and clarity of the MS.

-          Introduction is not up-to-date. Some references on bionanocomposites are missed (see for instance: Nanomaterials 2018, 8(11), 883; https://doi.org/10.3390/nano8110883; Nanomaterials. 7 (2017) 199–210. doi:10.3390/nano7080199)

-          Fig 4. Error bars should be added for a proper evaluation of the profiles.

-          The number of figures is very large. Some SEM can be moved to supplementary file. I would remove the figs with the size indication keeping the distributions in a single figure.

-          XRD. It appears that  110 signal is not proportional to the CNC content. I suggest the authors to put some efforts to get more detailed information from XRD.  

Author Response

Reviewer 2

Comments and Suggestions for Authors

The manuscript deals with polyvinylpyrrolidone/cellulose nanocrystals based nanocomposites from the preparation to characterization. The topic is interesting and there are new aspects worth of being published after revision mainly devoted to improve the impact and clarity of the MS.

-          Introduction is not up-to-date. Some references on bionanocomposites are missed (see for instance: Nanomaterials 2018, 8(11), 883; https://doi.org/10.3390/nano8110883; Nanomaterials. 7 (2017) 199–210. doi:10.3390/nano7080199) 

Authors' reply

Some up-to-date references (e.g., Nanomaterials 2018, 8, 883-897; Nanomaterials 2018, 8, 651-668) are added into Introduction section.

-          Fig 4. Error bars should be added for a proper evaluation of the profiles. 

Authors' reply

Error bars are added in Fig. 4 for a proper curves evaluation. 

-          The number of figures is very large. Some SEM can be moved to supplementary file. I would remove the figs with the size indication keeping the distributions in a single figure. 

Authors' reply

In the revised manuscript, the number of figures is reduced. Some figures are merged, some figures are moved to Supplementary Material.

-          XRD. It appears that  110 signal is not proportional to the CNC content. I suggest the authors to put some efforts to get more detailed information from XRD.  

Authors' reply

In the revised manuscript, the diffraction patterns are discussed in more detail.

Reviewer 3 Report

In this work authors report on the preparation of polyvinylpyrrolidone/CNC nanocomposite materials in the form of solid films and aerogels. Quite interesting insights are here reported and the preparation of PVP/CNC composites in films and aerogels provides an added value to the work. However, I find to major issues in this work. Firstly, I think authors prepared a rather descriptive work; I mean, they should try to find plausible explanations to observed properties in terms of different theoretical frameworks (see bellow provided suggestions). Moreover, the way the data is presented is not appropriate. Figure quality should be improved and I think the number of figures should be reduced to 8-10. After solving these issues, I think the paper could be accepted:

·         I suggest using the term “Cellulose nanocrystal” instead of “Nanocrystalline Cellulose” in the title as this is the generally accepted term.

·         The quality of Figure S1 is rather low. I wonder why one can see those spherical aggregates.

·         Figure 3 provides useful information on the water uptake behaviour of prepared nanocomposites. However, it is my opinion that authors do not properly discuss/analyze this part. I encourage them to further discuss this section and provide a reasonable explanation on obtained differences. Relevant works in the field: Journal of Materials Chemistry A 3 (2015) 13350-13356; Langmuir 31 (2015) 12170-12176; Polymer Engineering and Science 51 (2011) 2136-2142.

·         CNC incorporation markedly decreases the thermal stability of composites. The thermal stability of CNCs is particularly low. For instance, although the TGA curve corresponding to raw CNCs is similar to that shown in (Biomacromolecules 19 (2018) 2618−2628) the degradation occurs 60 °C earlier. Authors should compare these results, comment on that and provide an explanation about their lower thermal stability (which is a key parameter towards application).

·         I think DSC results are particularly interesting. A notable increase in Tg is achieved, which is typically explained in terms of macromolecular confinement provided by the CNC surfaces (Nano Letters 15 (2015) 6738-6744).

·         The curve quality in Figure 4 is rather poor. Authors can use the “smooth” tool in Origin to improve that.

·         A very notable increase on the Young´s modulus is obtained upon the CNC incorporation. To provide further light on that, authors should analyze their results in terms of Halpin-Tsai equation as done in Carbohydrate Polymers 142 (2016) 105–113.

·         I suggest removing Figure 5 into Supporting Information as currently it does not provide any valuable information.

·         A higher scale bar should be shown in Figures 6 and 7.

·         I suggest merging Figures 8-11 into one or two figures as now is quite tedious to have so many images.

·         I think that the last part of the manuscript is quite interesting. However, I think that providing so many figures makes its reading tedious. I suggest re-arranging this section, as it provides very useful information but currently it is not properly shown. Could authors envisage any specific application of such system? If authors could provide that, the quality of the work would be improved.

Author Response

Reviewer 3

Comments and Suggestions for Authors

In this work authors report on the preparation of polyvinylpyrrolidone/CNC nanocomposite materials in the form of solid films and aerogels. Quite interesting insights are here reported and the preparation of PVP/CNC composites in films and aerogels provides an added value to the work. However, I find to major issues in this work. Firstly, I think authors prepared a rather descriptive work; I mean, they should try to find plausible explanations to observed properties in terms of different theoretical frameworks (see bellow provided suggestions). Moreover, the way the data is presented is not appropriate. Figure quality should be improved and I think the number of figures should be reduced to 8-10. After solving these issues, I think the paper could be accepted:

         I suggest using the term “Cellulose nanocrystal” instead of “Nanocrystalline Cellulose” in the title as this is the generally accepted term.

Authors' reply

The article title is changed according to the reviewer's suggestion.

         The quality of Figure S1 is rather low. I wonder why one can see those spherical aggregates.

Authors' reply

The CNC particles produced by sulfuric acid hydrolysis have a rod-like morphology with an average length of 100-150 nm and width of 15-20 nm. It is obvious from analysis of TEM images.

         Figure 3 provides useful information on the water uptake behaviour of prepared nanocomposites. However, it is my opinion that authors do not properly discuss/analyze this part. I encourage them to further discuss this section and provide a reasonable explanation on obtained differences. Relevant works in the field: Journal of Materials Chemistry A 3 (2015) 13350-13356; Langmuir 31 (2015) 12170-12176; Polymer Engineering and Science 51 (2011) 2136-2142.

Authors' reply

In the revised manuscript, the experimental data in the section are discussed in more detail. Corresponding corrections are made to the manuscript. The relevant references are cited.

         CNC incorporation markedly decreases the thermal stability of composites. The thermal stability of CNCs is particularly low. For instance, although the TGA curve corresponding to raw CNCs is similar to that shown in (Biomacromolecules 19 (2018) 2618−2628) the degradation occurs 60 °C earlier. Authors should compare these results, comment on that and provide an explanation about their lower thermal stability (which is a key parameter towards application).

Authors' reply

Microcrystalline cellulose exhibits a higher thermal stability in comparison with CNC samples.  Because of the hydrolysis and presence of sulfate groups, the values of Ton for the first degradation stage for CNC sharply decrease to about 160°C compared with microcrystalline cellulose. Besides sulfate groups, the introduction of sodium on the surface of CNC may be another factor that significantly affects their thermal stability. If the acid sulfate groups of CNC surface are neutralized by NaOH solution, the values of Ton of thermal degradation considerably shift to a higher temperature (Nanoscale, 2014, 6, 5384–5393; Composites Communications 2 (2016) 15–18).

In the cited paper (Biomacromolecules 19 (2018) 2618−2628), the authors used for composites preparation the CNC suspension neutralized by NaOH (ζ-potential is -0.60 V). In our work we use acid CNC suspension (pH 2.4) (with a high content of sulfate groups, high surface charge and a high colloidal stability, ζ-potential is about -60 mV). That is why one can observe the difference in thermal stability.

         I think DSC results are particularly interesting. A notable increase in Tg is achieved, which is typically explained in terms of macromolecular confinement provided by the CNC surfaces (Nano Letters 15 (2015) 6738-6744).

Authors' reply

These considerations are included in discussion. The reference is included in the references list.

         The curve quality in Figure 4 is rather poor. Authors can use the “smooth” tool in Origin to improve that.

Authors' reply

Spline algorithm was used to improve the curves quality. Error bars are added for a proper curves evaluation.

         A very notable increase on the Young´s modulus is obtained upon the CNC incorporation. To provide further light on that, authors should analyze their results in terms of Halpin-Tsai equation as done in Carbohydrate  Polymers 142 (2016) 105–113.

Authors' reply

In the revised manuscript, the tensile properties of the PVP/CNC composite films are discussed in more detail, particularly in terms of Halpin-Tsai model. Corresponding changes and additions are made to the manuscript.

         I suggest removing Figure 5 into Supporting Information as currently it does not provide any valuable information.

Authors' reply

In concordance with a suggestion of Reviewer 1, the powder diffraction patterns are discussed in more detail.

         A higher scale bar should be shown in Figures 6 and 7.

Authors' reply

A larger scale bar is used in the figures.

         I suggest merging Figures 8-11 into one or two figures as now is quite tedious to have so many images.

Authors' reply

In the revised manuscript, the number of figures is reduced. Some figures are merged, some figures are moved to Supplementary Material.

         I think that the last part of the manuscript is quite interesting. However, I think that providing so many figures makes its reading tedious. I suggest re-arranging this section, as it provides very useful information but currently it is not properly shown. Could authors envisage any specific application of such system? If authors could provide that, the quality of the work would be improved.

Authors' reply

This section is rearranged. Photos of redispersed suspensions are moved to Supplementary Material. The authors attempted to provide a specific application area for such systems.

Round  2

Reviewer 3 Report

The authors have done a notable effort in order to improve their manuscript. New data has been added and the discussion has been markedly improved. Now the work provides many interesting insights on cellulose composite materials, which may be of interest for the scientific community. Accordingly, I suggest accepting the paper for publication in its current form. In any case, I will also like to raise awareness on the fact that for future works, I think authors should try to show their data in a more clear way as currently the large amount of figures results in a slightly complex reading.